# Pilot Study of a Novel First-Line Protocol (THOP) for Intermediate–Large B-Cell Lymphoma in Dogs

**DOI:** 10.3390/vetsci12030251

**Published:** 2025-03-06

**Authors:** Alejandra Tellez Silva, Ester Yang, Marlie Nightengale, Nikolaos Dervisis, Shawna Klahn

**Affiliations:** 1Department of Small Animal Clinical Sciences, Virginia-Maryland College of Veterinary Medicine, Virginia Tech, 215 Duck Pond Dr., Blacksburg, VA 24061, USAndervisi@purdue.edu (N.D.); 2Small Animal Clinical Sciences, University of Florida College of Veterinary Medicine, 2015 SW 16th Ave, Gainesville, FL 32608, USA; 3Department of Clinical Sciences & Advanced Medicine, University of Pennsylvania School of Veterinary Medicine, 3800 Spruce St., Philadelphia, PA 19104, USA; 4Veterinary Clinical Sciences, Purdue University College of Veterinary Medicine, 625 Harrison St., West Lafayette, IN 47907, USA

**Keywords:** dogs, canine, lymphoma, B-cell lymphoma, temozolomide, doxorubicin, THOP, chemotherapy

## Abstract

B-cell lymphoma is one of the most common cancer diagnoses in pet dogs. Currently, the standard treatment is a 19- or 25-week chemotherapy protocol consisting of four chemotherapy drugs: doxorubicin, cyclophosphamide, vincristine, and prednisone, or CHOP. CHOP is very effective, but half of dogs do not finish the treatment because the lymphoma relapses, the timing of which has been associated with cyclophosphamide. The goal of this prospective clinical trial was to identify side effects and early indication of effectiveness of a protocol that replaces cyclophosphamide with a different chemotherapy drug, temozolomide, to create a new protocol, THOP, that is only 15 weeks long. Fourteen pet dogs with B-cell lymphoma were treated with THOP as their first chemotherapy protocol. All dogs finished the treatment in complete remission. The results indicate that the protocol is well tolerated and may have similar effectiveness to the current standard of care. These data will be helpful in designing a larger prospective clinical trial that directly compares THOP with CHOP effectiveness.

## 1. Introduction

Lymphoma is one of the most common cancers diagnosed in dogs. It comprises 7–24% of all canine neoplasia and 83% of all canine hematopoietic malignancies. The most common form of lymphoma in dogs is diffuse large B-cell lymphoma, a type of multicentric lymphoma [1]. This disease is exquisitely chemosensitive but rarely cured. The chemotherapy protocol associated with the best outcomes is a multi-agent chemotherapy protocol consisting of cyclophosphamide, doxorubicin, vincristine, and prednisone (CHOP), with each drug administered as a single agent on a rotating basis over 19 or 25 weeks [2,3,4]. The reported response rate in these studies is 80–100%, with a progression-free survival (PFS) of 7–8 months. Attempts to reduce the duration of the CHOP protocol to 15 weeks has yielded similar response rates, albeit with shorter remission durations [5,6].

Unfortunately, not all dogs starting treatment with a CHOP protocol complete it in remission. Recently, it was reported that approximately half of dogs do not complete the full course of treatment, with progression of disease while on-protocol as the cause in 70% of cases [2]. Progression during or shortly after completion of CHOP is negatively associated with treatment response and patient outcomes during rescue chemotherapy [7]. Relapse on-protocol with CHOP as first-line treatment has been temporally associated with the administration of cyclophosphamide, which suggests that its antineoplastic activity may be lower compared to other cytotoxic drugs in the same protocol.

Cyclophosphamide is an alkylator that requires metabolic activation but is unaffected by the pleiotropic-glycoprotein drug efflux pump (PGP-pump). However, other resistance mechanisms such as increased expression of O6-alkyl guanine DNA alkyl transferase have been proposed as cellular resistance mechanism [6]. Temozolomide (TMZ) is a bifunctional oral alkylator. Its metabolism and mechanism of action are similar to dacarbazine but does not require hepatic metabolic activation by cytochrome P450. TMZ is converted to the active form MITC 5-(3-methytriazen-2-yl) imidazole-4-carboxamide, breaking down to 5-aminoimidazole-4-carboxamide (AIC) and the reactive methyl diazonium ion. This reaction causes the formation of O6 methylguanine which is believed to be responsive for its cytotoxic effect [8]. There are few reports of the usage of TMZ in dogs with cancer, and only recently has the maximally tolerated dose (MTD) as a sole agent in dogs been established [9]. The MTD was determined to be 150 mg/m^2^ orally, daily for five consecutive days, repeated once every four weeks. The dose-limiting toxicities (DLT) noted were hepatotoxicity and thrombocytopenia, with gastrointestinal toxicity also frequently noted [9]. This raises concerns of overlapping toxicity if administered in combination with doxorubicin, despite a promising response rate of 60–70% in dogs with relapsed and refractory lymphoma [10,11].

Our group sought to overcome the limitations of the most widely employed CHOP protocols, specifically, the length of treatment and the high on-protocol failure rate. Our rationale was that the substitution of cyclophosphamide with temozolomide in a chemotherapy protocol of doxorubicin, vincristine, and prednisone (THOP) would be a rational strategy as a first-line treatment for canine intermediate–large cell B-cell lymphoma. Lack of outcome data in the treatment of naïve canine lymphoma patients combined with ethical concerns regarding the potential for significant toxicity led to the development of a prospective, single arm, pilot study. The objectives were to describe the adverse event profile and to generate preliminary evidence of efficacy of THOP as first-line treatment in dogs with treatment-naïve intermediate–large B cell lymphoma. The data generated from this pilot study will be used to design an appropriately powered, randomized phase III clinical trial to compare the effectiveness of THOP and the current standard of care of a CHOP-based protocol.

## 2. Materials and Methods

### 2.1. Ethical Statement

All clients were informed of the purpose of this study and informed consent was obtained. This study was approved by the Virginia Tech Institutional Animal Care and Use Committee (IACUC #21-118) and the Veterinary Hospital Board.

### 2.2. Patient Selection

Dogs diagnosed with treatment-naïve, multicentric, intermediate–large, B-cell lymphoma were recruited from the Animal Cancer Care and Research Center between August 2021 to August 2023. Dogs were eligible for inclusion with confirmation of diagnosis of intermediate–large lymphoma by cytology or histology [12,13]. The immunophenotype of B-cell lymphoma was determined by flow cytometry or immunohistochemistry, as previously described [3,14,15,16]. Patients must have had at least stage III disease to minimize potential variability in this study and to represent the most common clinical presentation. Dogs with stage V disease were eligible if the assignment to stage V was due to peripheral blood involvement. Dogs were required to be at least 1 year of age and 10 kg (22 lb) of weight. Dogs that received previous chemotherapy treatment were excluded, but corticosteroid treatment was acceptable if the duration was less than 10 days [17,18]. Exclusion criteria included patients with significant hematologic or organ dysfunction precluding treatment with chemotherapy, a diagnosis of any other type of lymphoma, or a diagnosis with stage V due to reasons other than peripheral blood involvement.

All patients were screened prior to enrollment with complete blood cell count and pathologist review, serum biochemistry, three-view thoracic radiographs, and abdominal ultrasound.

### 2.3. Chemotherapy Protocol

The THOP protocol consisted of five cycles of three weeks’ duration each, for a total planned treatment duration of 15 weeks (Table 1).

On week 1, vincristine was administered as an intravenous bolus at 0.7 mg/m^2^ and prednisone was administered orally at 40 mg/m^2^ per day for 7 days. On week 2, doxorubicin (DOX) was administered as a 20-min intravenous infusion at 30 mg/m^2^, in combination with TMZ orally at a target dose of 100 mg/m^2^ per day for five days. TMZ dose was based on prior studies of TMZ/DOX in dogs with lymphoma [10,11]. The first dose of TMZ was administered the same day as DOX administration, and owners were instructed to give the remaining four doses at home, starting the following day. Prophylactic antiemetic support was standardized for each TMZ administration: maropitant citrate (Cerenia) 2 mg/kg and ondansetron (Zofran) 0.5 mg/kg administered orally 1 h prior TMZ [11]. The owners were instructed to administer TMZ on an empty stomach and offer a meal 15–30 min after TMZ. On week 3, a physical exam and complete blood count were performed. All in-clinic treatments and week 3 visits were performed by the investigators.

### 2.4. Evaluation of Response and Monitoring

Lymph nodes selected as target lesions for disease response to therapy were selected based on The Veterinary Cooperative Oncology Group (VCOG) consensus document for response evaluation criteria for peripheral nodal lymphoma in dogs (v1.0) and on lymph node accessibility for reliable caliper measurements [19]. A complete response (CR) was defined as disappearance of all evidence of disease. All lymph nodes had to be non-pathologic in size in the judgement of the evaluators. A partial response (PR) was defined as a decrease of 30% in the mean sum of the longest diameter (LD) of the lymph node size. Progressive disease (PD) was defined as at least a 20% increase in the mean sum LD taking as reference the smallest mean sum LD at baseline or during follow-up. The LD of at least one of the target lesions must demonstrate an absolute increase of at least 5 mm compared with its nadir for PD to be defined. Stable disease (SD) was defined as neither 30% decrease nor 20% increase [19].

All patients were re-staged with 3-view thoracic radiographs and abdominal ultrasound at week 6 or 7 of the protocol. Any new or unresolved liver or spleen lesions identified at restaging were evaluated with fine needle aspirate cytology. After completion of the 15-week THOP protocol, active surveillance was performed with monthly physical examinations and complete blood cell count with pathologist review performed for a minimum of 1 year after the protocol was completed. Any relapses were determined by physical exam by the investigators and confirmed with lymph node fine needle aspirate and cytology. All response and monitoring visits were performed by the investigators.

### 2.5. Toxicity

Veterinary Cooperative Oncology Group—Common Terminology Criteria for Adverse Events (VCOG-CTCAE v2) following investigational therapy in dogs and cats was used to assess and grade gastrointestinal, hematologic, or other toxicities [20]. Hematologic adverse events were determined by review of complete blood counts performed weekly during the treatment protocol; constitutional or gastrointestinal adverse events were determined from the owner history reported at each patient visit.

### 2.6. Statistical Analysis

Time to best response (TBR) was defined as the time from THOP initiation to the time of complete resolution of target lesions. Response duration (RD) was defined as the time from the best response to the time of relapse for dogs that achieved CR or PR. Time to progression (TTP) was defined as the time from the THOP initiation to the time of relapse. Overall survival (OS) was defined as the time from the THOP initiation to the time of death due to lymphoma. The Kaplan–Meier survival analysis method was used to estimate TBR, RD, TTP, and OS curves following treatment. The log-rank test was used to compare Kaplan–Meier curves between different potential risk factor groups (stage, substage, sex, neuter status, and response to treatment). The Cox proportional hazards regression method was used to determine whether the aforementioned potential risk factors were associated with TBR, RD, TTP, and OS. Descriptive statistics were used to describe the study population characteristics, toxicity profile of the THOP protocol, and achieved TMZ dose.

All reported *p* values are 2 sided. Values of *p* < 0.05 were considered significant. Statistical analyses were performed with standard software (MedCalc^®^ Statistical Software version 22.030; MedCalc Software Ltd., Ostend, Belgium; https://www.medcalc.org; accessed on 1 March 2024)

## 3. Results

### 3.1. Study Population

Twenty dogs were intended as the cohort size for this pilot study. A total of 14 of the 20 planned dogs were enrolled due to unanticipated recruitment challenges. All 14 enrolled dogs were treated with the 15-week THOP chemotherapy protocol at ACCRC between September 2021 and May 2023. The mean age was 8.0 years (SD = 3.4). Seven dogs were spayed females, six dogs neutered males, and one dog was an intact male. The mean weight was 25.4 kg (SD = 7.5). There were five different breeds represented, the most common being mixed breed dogs, followed by Pitbull and one of each breed: Goldendoodle, Beagle and Rottweiler.

Full staging was performed in all dogs at study enrollment. Three dogs (21%) were stage III, five (35%) stage IV, and six (44%) stage V. Eight dogs (57%) were substage (a) and six (43%) substage (b).

Lymphoma was diagnosed via cytology in all 14 dogs, and one dog also had histopathology performed. Thirteen dogs were immunophenotyped with flow cytometry and one with immunohistochemistry. All dogs were confirmed to be diagnosed with intermediate–large B-cell lymphoma. Of the 13 dogs with flow cytometry information, 12 had high expression of MHCII and one had low expression of MHCII.

### 3.2. The THOP Protocol

All dogs (*n* = 14) completed the planned 15 weeks of treatment, with a total of 70 cycles of THOP chemotherapy administered. The time from initial diagnosis to initiation of THOP was 7.9 days (range: 0–15 days). The mean TMZ dose was 101.3 mg/m^2^ (SD: 7.3 mg/m^2^) and the cumulative dose per dog was 2533.6 mg/m^2^ (SD: 183.1 mg/m^2^).

### 3.3. Adverse Events

A total of 98 adverse events (AEs) were recorded for the 14 dogs included in this study. Approximately 10% of all AEs (*n* = 10) were classified as high-grade (grade III or IV), nearly all were hematological (Table 2).

Thirty-two (23%) AEs were classified as biochemical. All of them were either grade I or II, including 15/32 (46%) incidences of ALP elevation, 10/32 (31%) ALT elevation, and 7/32 (23%) BUN elevation. No interventions nor dose delays or reductions were required.

A total of 26 (26%) AEs were recorded as gastrointestinal, 25 of them were classified as grade I or II, including 17/25 (68%) inappetence events, 4/25 (16%) diarrhea, and 4/25 (16%) nausea. All were managed with symptomatic care and did not result in dose delays. However, TMZ dose reductions occurred due to owner request. One AE was grade III hematemesis, which was treated with intravenous fluids and injectable gastrointestinal medications for 48 h.

A total of 36 (37%) AEs were recorded as hematological, with 27 of the 36 (75%) classified as low grade (grade I or II), and 9/36 (25%) as high grade (grade III or IV). Anemia was recorded in 13/36 (36%) events, all of them grade I. Neutropenia was recorded in 20/36 (55%), including nine grade I, three grade II, four grade III, and four grade IV. Thrombocytopenia was recorded in 3/36 (8%) hematologic AEs, with two grade I and one grade III. None of these AEs resulted in febrile neutropenia or required patient hospitalization.

Of the nine high-grade hematological AEs, four were attributed to vincristine administration: three grade III and one grade IV episodes of neutropenia, resulting in 10 treatment delays and three vincristine dose reductions. The remaining five high-grade hematological AEs were attributed to the DOX/TMZ combination: one grade III neutropenia, three grade IV neutropenias, and one grade III thrombocytopenia. These AEs resulted in five treatment delays, seven DOX dose reductions and four TMZ dose reductions due to persistent low-grade GI adverse events (nausea, vomiting, and decreased appetite) (Table 3).

Other recorded AEs included single incidents of grade II alopecia, grade II hyperpigmentation, grade II urinary incontinence, and grade I foot ulceration. None of these events required any intervention.

### 3.4. Treatment Response and Survival

The overall response rate in this population of dogs was 100%, with all patients achieving CR. All patient outcome data were included in the analysis. The median time to best response (TBR) of 26.5 days (95% CI: 21.9–31.1) (Figure 1). The median response duration (RD) was 269 days (95% CI: 259–414) (Figure 2). The median time to progression (TTP) was 291 days (95% CI: 280–439) (Figure 3), and the median OS achieved was 433 days (95% CI: 335–499) (Figure 4). No treatment differences were noted between sexes with regard to time to progression (*p* = 0.9) or overall survival (*p* = 0.7).

During the study period, 10 of the 14 dogs experienced relapse, with 6 receiving at least one rescue protocol (range: 1–3), including LOPP, MOPP, rabacfosadine, lomustine, and L-asparaginase. One dog resumed THOP as a rescue regimen. Four dogs did not receive rescue chemotherapy following relapse. Two dogs were euthanized while in CR due to causes unrelated to lymphoma at 173 and 232 days post-THOP initiation. At the time of data analysis, two dogs remained alive and in CR at 392 and 464 days post-THOP initiation.

## 4. Discussion

The objectives of this pilot study were to evaluate the safety, feasibility, and preliminary efficacy of a novel multi-agent, cyclophosphamide-free, doxorubicin-based chemotherapy protocol (THOP) as a first-line treatment for canine intermediate–large B-cell lymphoma. Our findings suggest that THOP is a well-tolerated protocol, offering a shorter treatment duration than current standard-of-care protocols while achieving durable remissions. The majority of the dogs in this study achieved CR by the start of the second cycle (week 4 of the protocol). The overall response rate (ORR), time to progression (TTP), and overall survival (OS) were 100%, 291 days, and 433 days, respectively, which is comparable to 19- or 25-week CHOP-based protocols [2,3,4], although the OS should be interpreted with caution due to the use of rescue chemotherapy by 60% of the relapsed dogs.

The toxicity profile of THOP was acceptable without evidence of significant overlapping DOX/TMZ toxicities, and the biochemical adverse events (AEs) in this study were all low-grade. Increased liver enzyme values were attributed to prednisone administration or preexisting lymphoma infiltration rather than hepatotoxicity secondary to TMZ, as the values returned to within reference range without intervention once patients achieved CR status. Hematological AEs accounted for the majority (9/10) of high-grade AEs. High-grade neutropenia comprised 25% of the hematological AEs; however, all cases were afebrile and none required hospitalization. High-grade thrombocytopenia was not a significant hematological AE in this study. Gastrointestinal AEs were predominantly low grade and self-limiting, with all but one resolving within three days with supportive therapy. Over the 140 treatment weeks, only 15 treatment delays were necessary, which were primarily following vincristine administration. Dose reductions in vincristine and DOX were exclusively related to neutropenia, while TMZ dose reductions occurred at the owners’ request due to low-grade but persistent gastrointestinal symptoms, notably nausea—a well-documented side effect in human and veterinary oncology [11,21,22,23].

The cornerstone of lymphoma treatment is a multidrug protocol, with anthracyclines often providing extended remission durations. However, at relapse, cumulative drug toxicity and resistance limit the available rescue options. The data generated by this pilot study suggest that the THOP protocol can provide comparable response and survival outcomes to CHOP-based protocols while requiring fewer clinic visits (10 visits) and a shorter overall treatment duration (15 weeks), thereby offering a potential advantage over the current standard of care. Additionally, all enrolled dogs completed the intended treatment duration of 15 weeks. A randomized phase III clinical trial is needed to confirm these promising results.

The use of TMZ in veterinary medicine has historically been limited by cost and availability [10,11,24,25,26,27,28]. Recent improvements in accessibility and affordability have made its incorporation into protocols like THOP feasible. In our practice, the weekly cost of THOP was comparable to that of standard CHOP-based protocols, and its shorter duration offers a potential to decrease the required financial investment for families without compromising patient outcome.

A key consideration highlighted by this study is the importance of reinforcing owner compliance with antiemetic administration during TMZ-containing weeks. While TMZ-induced nausea and inappetence are not life-threatening, they can cause significant stress to owners and risk early protocol discontinuation [11].

A notable strength of this study was the standardized active surveillance protocol following treatment completion. Monthly follow-ups, including clinical examinations and routine lab work, facilitated consistent relapse detection, bolstering confidence in the reported outcomes. This level of surveillance should be integrated into future clinical trial designs. Our study’s follow-up protocol did not include restaging with thoracic radiographs and abdominal ultrasound, thus it is possible that a number of dogs may have had disease relapse prior to their classification as PD based on physical examination.

This pilot study is not intended to provide definitive conclusions but rather to generate data for powering randomized trials. Although results are promising, the small sample size and potential biases, such as the inclusion of treatment-naïve patients with intermediate–large B-cell lymphoma and high MHCII expression, may limit generalizability. In addition, all but one dog was spayed or neutered, which may have limited our ability to discern any sex-related differences in treatment effectiveness. Nevertheless, these findings provide critical insights for designing future prospective trials comparing THOP with CHOP-based protocols.

## 5. Conclusions

THOP appears to be a safe, well-tolerated, and effective first-line treatment for canine intermediate–large B-cell lymphoma. Prophylactic antiemetics are essential to mitigate gastrointestinal side effects associated with DOX/TMZ. Future research should focus on randomized clinical trials to validate these findings and further assess the comparative efficacy of THOP- and CHOP-based protocols.

## Figures and Tables

**Figure 1 vetsci-12-00251-f001:**
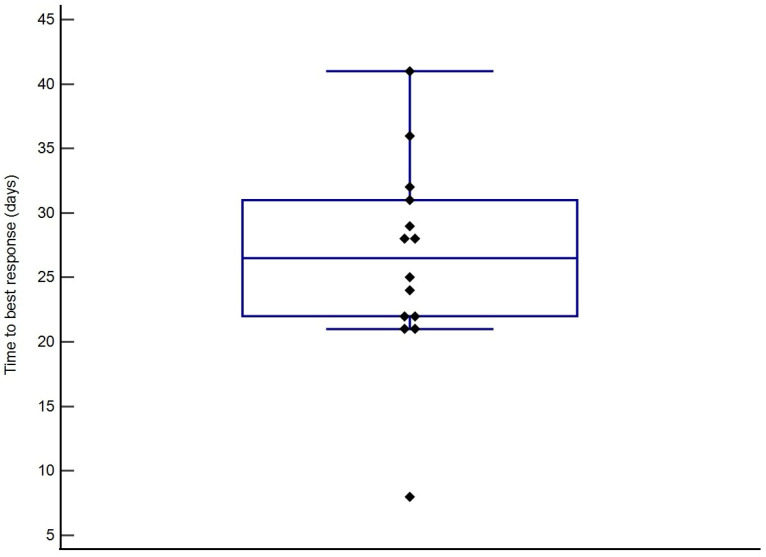
Box plot showing best time to response (BTR). The majority of patients presented best response after completion of the first THOP cycle.

**Figure 2 vetsci-12-00251-f002:**
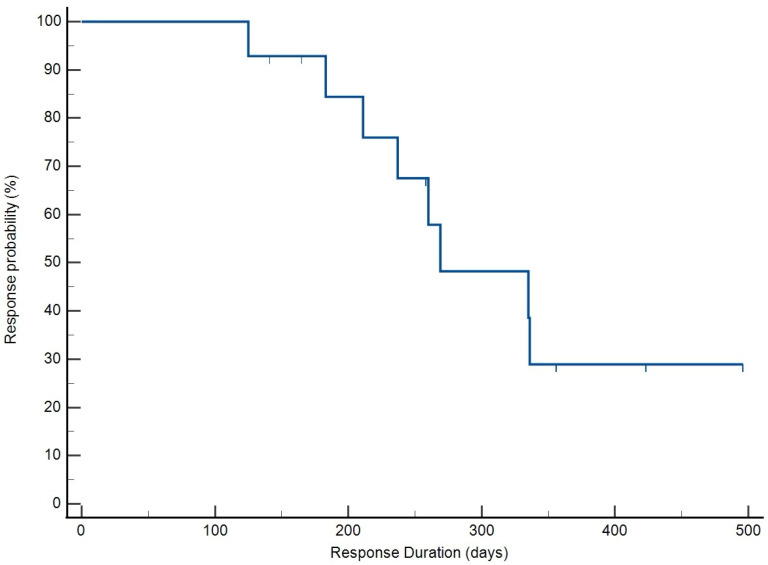
Kaplan–Meier survival curve illustrating response duration on patients receiving THOP. Median response duration was 269 days.

**Figure 3 vetsci-12-00251-f003:**
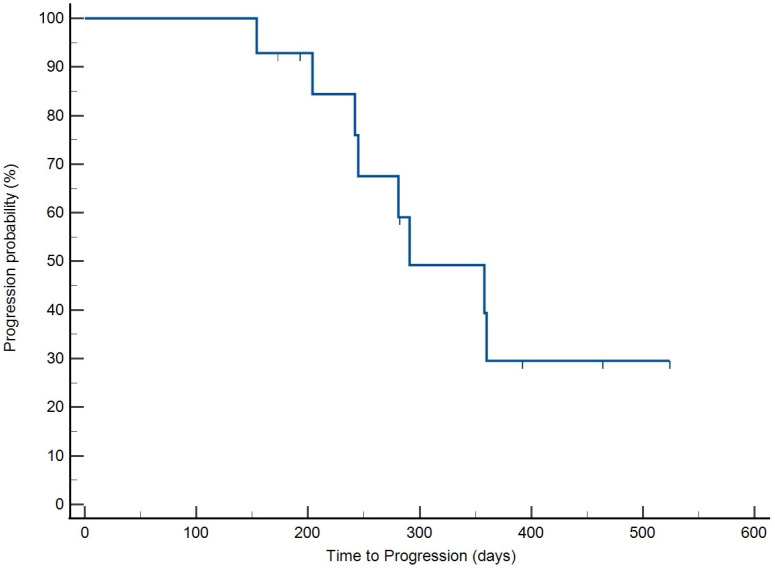
Kaplan–Meier survival curve presenting the time to progression for patients receiving THOP. Median time to progression was 291 days.

**Figure 4 vetsci-12-00251-f004:**
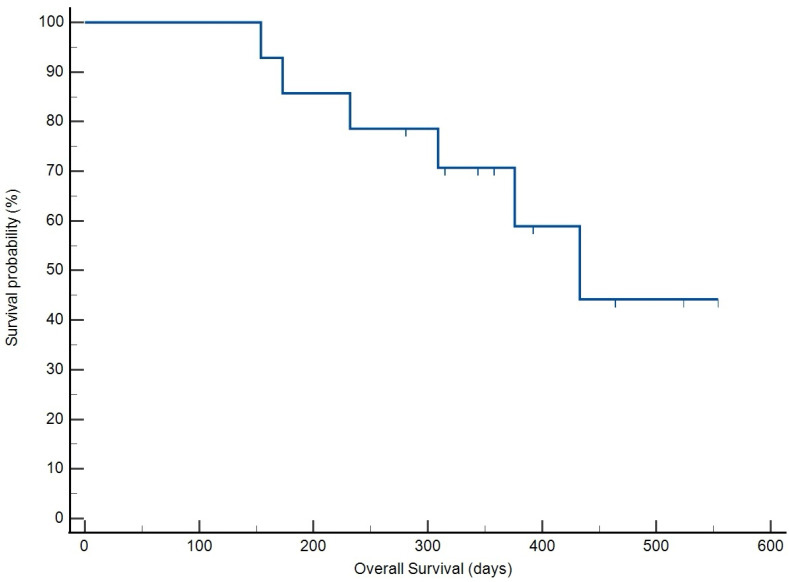
Kaplan–Meier survival curve illustrating overall survival of patients receiving THOP treatment. Median overall survival was 433 days, including patients receiving rescue protocols (*n* = 14).

**Table 1 vetsci-12-00251-t001:** Screening, restaging, and 15-week THOP protocol (temozolomide, doxorubicin, vincristine, and prednisone). “X” denotes that the procedure occurs during the specific week of the protocol.

	Week	Screening	1	2	3	4	5	6	7	8	9	10	11	12	13	14	15
Procedure																	
**Vincristine 0.7 mg/m^2^ IV**			X			X			X			X			X		
**Prednisone 40 mg/m^2^ PO q24 h for 7 days**			X			X			X			X			X		
**Doxorubicin 30 mg/m^2^ IV**				X			X			X			X			X	
**Temozolomide 100 mg/m^2^ PO q24 for 5 days**				X			X			X			X			X	
**CBC**		X	X	X	X	X	X	X	X	X	X	X	X	X	X	X	X
**Chemistry**		X		X			X			X			X			X	
**Pathologist review of CBC**		X						X									
**Urinalysis**		X															
**Cytology or histopathology**		X															
**Flow cytometry or immunohistochemistry**		X															
**Thoracic radiographs**		X						X									
**Abdominal ultrasound**		X						X									

**Table 2 vetsci-12-00251-t002:** Adverse events recorded during THOP administration, graded according to Veterinary Cooperative Oncology Group—Common Terminology Criteria for Adverse Events (VCOG-CTCAE v2) following investigational therapy in dogs and cats. A total of 10% of AEs were grade III–IV, with neutropenia as the most common high-grade AE, and all were non-febrile.

	Grade I	Grade II	Grade III	Grade IV
**Neutropenia**	9	3	4	4
**Anemia**	13			
**Thrombocytopenia**	2		1	
**ALP elevation**	13		1	1
**ALT elevation**	7	3		
**BUN elevation**	5	2		
**Nausea**	2	2	1	
**Inappetence**	11	6		
**Diarrhea**	1	3		

**Table 3 vetsci-12-00251-t003:** High-grade hematological adverse events attributable to doxorubicin/temozolomide or vincristine administration and the resulting treatment modifications (delay or subsequent dose reductions).

Drug	Neutropenia	Thrombocytopenia	Treatment Delays	Dose Reductions
**Vincristine**	Grade III: 3 Grade IV:1	None	10	3
**Doxorubicin/Temozolomide**	Grade III: 1 Grade IV:3	Grade III: 1	5	Doxorubicin: 7 Temozolomide: 4

## Data Availability

All generated or analyzed data during this study are included in this article. The datasets used for analysis are available from the corresponding author upon request.

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
