# Peer review of "Pilot Study of a Novel First-Line Protocol (THOP) for Intermediate–Large B-Cell Lymphoma in Dogs"

_vetsci, 2025, doi:10.3390/vetsci12030251_

Round 1

Reviewer 1 Report

Comments and Suggestions for Authors

The work herein submitted  "Pilot Study of a Novel First‐Line Protocol (THOP) for Intermediate‐Large B‐cell Lymphoma in Dogs" is well structured and clear to read. It is original and has scientific interest for the medical veterinary community.

It has some limitations in methodology, namely in the way response to treatment and tumor progression was assessed that strongly impacts the statements and data for the evaluated end-points: TBR, TTP and OS.

Here are some comments/ suggestions to help improving its scientific content:

Lines 142-143 : How stage 4 disease was confirmed? How was liver and spleen infiltration assessed at baseline and in the follow up ( re-staging?) If not performed, authors should state how  stage 4 diagnosis was performed or if it was presumptive, and discuss the associated limitations  in assessing response to treatment.

Lines 159-160: How can authors be sure of TBR and TTP  when the first  re-staging was done only at week 6 or 7 . All the response or disease response and progression was missed without additional exams ( namely US and Chest XR). Please include this limitation and discuss the impact in evaluation of treatment response.

Line 247 : 60% of dogs that relapsed during treatment received rescue therapy , consequently increasing survival. Please discuss this subject and its impact in OS.

Lines 273-274: How can authors affirm the dogs were in complete remission without abdominal US and X-rays? The re-staging was only done at week 6/7. Remission status  and TBR cannot be determined at week 4! Please discuss.

 Lines 312-314 : Authors state that monthly follow‐ups, including clinical examinations  routine lab work, facilitated consistent relapse detection, bolstering confidence in the reported outcomes. This level of surveillance should be integrated into future clinical trial designs. How can authors be confident and advise monthly follow ups  if more detailed exams like chest X-rays and abdominal ultrasound were not performed? This has strong limitation in detecting internal disease progression. These sentence should be rewritten and include the limitations of a monthly follow up without US and  chest X-rays.

Reviewer 2 Report

Comments and Suggestions for Authors

The article ‘Pilot Study of a Novel First‐Line Protocol (THOP) for Intermediate‐Large B‐cell Lymphoma in Dogs’ is a well written manuscript and provides  interesting information on the application of a 'possibly' new treatment (THOP) for B cell lymphoma in dogs.

In this reviewer’s opinion, however, some comments should be addressed prior to publication.

Line 50: Please add an introductory sentence to help the reader understand the main point of discussion within the paragraph.

Line 99:  Please added  reference(s) for protocols used for immunohistochemistry and flow cytometry analysis.

Line 100: Please explain why Stage III of the disease was the criteria (threshold) for acceptance of dogs within the study.

 Line 133: Please identify which lymph nodes (ie popliteal LN?) were assessed for the presence of disease or treatment efficacy.

In the results section. Please provide a table or information as written text - with associated comments - showing treatment differences ( if any) between male and females dogs.

 In the discussion (associated with the above comment).  Please comment if the sex of the dog could affect treatment outcomes. In people- men often have increased risk and poor survival rates in cancers- including lymphoma. Was this observed in your study?
